# The Application of UVC Used in Synergy with Surface Material to Prevent Marine Biofouling

**Kailey N. Richard \*, Kelli Z. Hunsucker, Harrison Gardner, Kris Hickman and Geoffrey Swain**

Center for Corrosion and Biofouling Control, Florida Institute of Technology, Melbourne, FL 32901, USA; khunsucker@fit.edu (K.Z.H.); hgardner2009@my.fit.edu (H.G.); khickman2019@my.fit.edu (K.H.); swain@fit.edu (G.S.)
**\*** Correspondence: krichard2018@my.fit.edu

**Abstract:** Biofouling is problematic for the shipping industry and can lead to functional and financial setbacks. One possible means of biofouling prevention is the use of ultraviolet-C (UVC) light. Previous studies have investigated UVC with marine coatings, but the synergistic effect with color and surface material, specifically reflectance, has yet to be determined. This study comprised three parts: UVC and color (red vs. white), UVC and reflectance (stainless steel vs. polycarbonate), and UVC and exposure intervals (weekly intervals and 10 min intervals). There was no variance in the biofouling communities for colored surfaces when exposed to 254 nm UVC. Reflectance studies demonstrated that the surface material plays a role in biofouling settlement. Stainless steel panels had significantly greater macrofouling settlement than polycarbonate, specifically among encrusting bryozoan, tubeworms, and tunicate communities. Panels of both surface materials exposed to indirect UVC significantly differed from controls and those exposed directly to UVC. Exposure intervals were also found to reduce biofouling settlement especially with long frequent intervals (i.e., 10 min/day). UVC can be utilized on various colored surfaces and different surface types, but the effectiveness in preventing biofouling is ultimately determined by the duration and frequency of UVC exposure.

**Keywords:** ultraviolet light; biofouling; color; surface material; exposure interval





## 1. Introduction

Biofouling is a significant problem for the shipping industry as it leads to many functional and financial setbacks. During idle periods, a ship can accumulate biofouling. Within minutes of immersion, a surface, such as a ship hull, accumulates a conditioning film, followed by bacteria and other unicellular organisms (i.e., diatoms). This accumulation is known as slime or microfouling [1]. Within hours to days, macrofouling communities such as barnacles, sponges, and tunicates may begin to grow, creating a complex biofouling community. This accumulation causes changes in surface roughness that lead to increased drag, lower fuel efficiency, and increased greenhouse gas emissions [2]. A ship with only biofilm (microfouling) and no macrofouling can cause up to $1.2 million in fuel consumption [2]. Significant growth on a ship hull may also require physical removal (e.g., brushing, power washing, scrapping) of the biofouling organisms [2]. Biofouling can also damage oceanographic equipment and is a known vector for the transfer of invasive species [3–5].

Currently, the most common approach to biofouling prevention on ship hulls is the application of marine coatings. These fall into two main categories: antifouling (AF) and fouling release (FR) [6]. Antifouling coatings are biocide-based, such as copper, which deters growth, making it hard for other organisms to attach and develop [1,6,7]. The drawback is these coatings emit copper and other chemicals into the water, causing environmental issues [8] and potentially resulting in nutrient loading [7]. Fouling release coatings have been developed as an alternative to biocide coatings. These coatings are often silicone-based and work by reducing the adhesion strength of organisms, allowing for easier

removal either via cleaning or hydrodynamic forces imparted on the ship hull as it moves through water [1,6,9]. Other approaches include mechanical methods which either prevent the buildup of biofouling or remove a fully established biofouling community. Grooming, which uses remotely operated vehicles (ROVs) equipped with brushes to frequently and gently wipe the ship hull, can remove fouling before it has time to fully establish [10,11]. Grooming at a frequency of once a week has been effective in removing micro- and macrofouling on both AF and FR coatings [10–13]. Cleaning is another alternative, which is a reactive approach to fouling control, as it aims to remove growth via power washing either in or out of water. Currently, there are concerns that grooming and cleaning may release coating biocidal materials into the water; thus, there are methods in place to capture the effluent [11,14]. While there are many types of antifouling methods and strategies available, there is still a need for solutions which are environmentally friendly [12].

Ultraviolet-C (UVC) light is commonly used for the prevention of bacteria in the medical field. For medicinal purposes, it has proven to be 99% effective in eliminating bacterial biofilms growing on catheters. UVC light (254 nm) damages bacterial cells by attacking their DNA [15]. Recently, it has been applied in the marine field to prevent biofouling formation on multiple surfaces due to its increased affordability, and it is also considered an environmentally friendly method of biofouling removal or prevention [16–23]. UV radiation can disrupt the detection and settlement of coral larvae along with decreasing the biofilm formation on various surfaces [16–23]. Barnacles have also been shown to undergo periods of blindness when exposed to UVB light, preventing them from detecting surfaces and settling [21]. Salters and Piola (2017) [17] used embedded UVC LEDs as a way to prevent biofouling and identified no biofouling within a small vicinity of the light source.

Recent experiments have investigated the use of UVC as an externally applied source for biofouling prevention on ship hulls. Hunsucker et al. (2019) [22] and Braga et al. (2020) [23] determined that biofouling was prevented or significantly decreased in abundance when various frequencies of UVC were applied. Both experiments proved UVC can work in synergy with the fouling control coatings (biocidal and fouling release) mentioned above. On the basis of these prior experiments, a three-part study was designed to address how UVC can be used in synergy with different surfaces while considering the importance of exposure frequency. UVC has yet to be applied to surfaces with varying reflective properties or colors that are unattractive to biofouling organisms. The use of inert surfaces removes the use of antifouling or fouling release coatings, in turn removing the adverse effects that these coatings could potentially have on the environment. Additionally, the use of color is known to influence biofouling settlement and recruitment. Swain et al. (2006) [24] found that white surfaces contained less biofouling than black surfaces. Red-colored surfaces are a highly preferred color for settlement of barnacles including *Balanus amphitrite*, tubeworms, and some coral species [25–29]. Colors reflect light at different frequencies, and it is thought that color would influence settlement during periods when there is no UVC exposure.

The aim of this study was to investigate the interaction of UVC on surfaces with different reflective properties and color, as well as its effect on biofouling settlement. Additionally, only daily exposure intervals (1 min/day and 1 min/6 h) were tested in previous studies. Thus, study examines various exposure times and intervals (e.g., 5 min/week and 10 min twice a week) in order to prevent biofouling formation.

## 2. Materials and Methods

A three-part study was designed to address how UVC interacts with surfaces. Specifically, Section 2.1 looked at how surface color and UVC exposure work in synergy to prevent biofouling settlement, while Section 2.2 compared the interaction of UVC light with surfaces of different reflective properties (stainless steel and polycarbonate). In Sections 2.1 and 2.2, low doses of UVC known to have a limited effect on fouling [22,23], were applied to panels to allow for a comparison of fouling intensity and composition. Section 2.3 investigated



the effectiveness of different time intervals to prevent biofouling formation on the surfaces used in Sections 2.1 and 2.2.

### 2.1. UVC and Color

To test the synergistic effect of UVC with color, red and white surfaces were selected according to prior research on biofouling settlement [25–29]. Both colors were printed on weatherproof computer paper. Each colored paper was sandwiched between 10 cm × 20 cm × 0.16 cm polycarbonate plates to create a uniform surface (herein referred to as a panel). The polycarbonate panels were 1.6 mm thick to allow for the color paper to be clearly visible. General Electric (GE) all-purpose silicone was used to seal the edges of panels to prevent water ingression which would alter the color and paper. Trilux-33 (Interlux), an antifouling spray paint, was used as a border on the panels to prevent the edge effect of unwanted biofouling.

Eight polycarbonate panels were housed on two sides of a 3D printed box (referred to as Box 1), which also allowed water and biofouling larvae to flow through [22] (Figure 1). A 25 W Aqua UVC (254 nm) lamp was placed in the center, at a distance of 25 mm from the panels [22,23]. In fresh water, the light intensity of the UVC lamp 25 mm from a surface was measured at $1.31 \pm 0.88$ $\mu W/cm^2$. Four replicates of red polycarbonate panels and four white polycarbonate panels were placed in Box 1 (Table 1). A PRIME-digital lighting timer was used to set to a time frequency of 1 min per day within the box [22,23]. Control panels were constructed in the same fashion and were hung on a PVC frame with no UVC exposure.

### 2.2. UVC and Reflectance

An experiment was designed to determine how fouling interacted with surfaces that reflect UVC or absorb it. Substrates of stainless steel and polycarbonate were used because they are both inert and have known specular reflectance at 254 nm. Stainless steel reflects 55% [30] and polycarbonate reflects 0% [31] of UVC light at near-normal angles of incidence. The reflectivity of the two substrates was measured in the lab using the same lamp as seen in Figure 1 and a Solar Light (model no. PMA2122-WP) UVC sensor. Laboratory testing confirmed that stainless steel panels reflected significantly more UVC light than the polycarbonate panels.

In order to test the interaction of UVC with surfaces of different reflectance, three different treatments were established (Table 1): all polycarbonate panels (Box 1), all stainless steel panels (Box 2), and a combination of polycarbonate panels and stainless steel panels (Box 3). Note that the box setup used to investigate the synergistic effect of UVC and color was also Box 1. Eight replicates of 316 stainless steel, polished to an ASTM 480 # 8 mirror finish [32], were cut into 10 cm × 20 cm panels and placed in Box 2. Box 3 contained four stainless steel panels on one side of the box and four red polycarbonate panels on the opposite side of the box, to test for the indirect effect of the reflective stainless steel on polycarbonate. Each box was set to a time frequency of 1 min per day using a PRIME-digital lighting timer based on prior experiments [22,23]. A set of control panels was constructed for each surface material, immersed during the same time period, and not exposed to UVC.

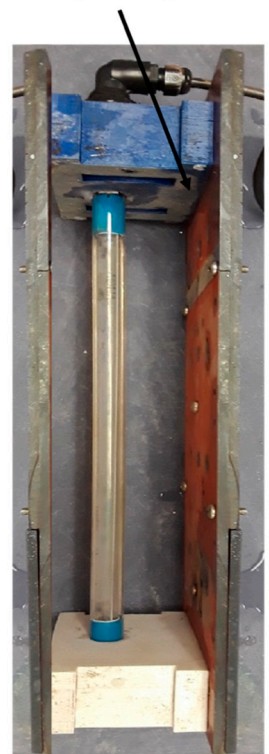

UVC lamp is positioned 25 mm away from panels

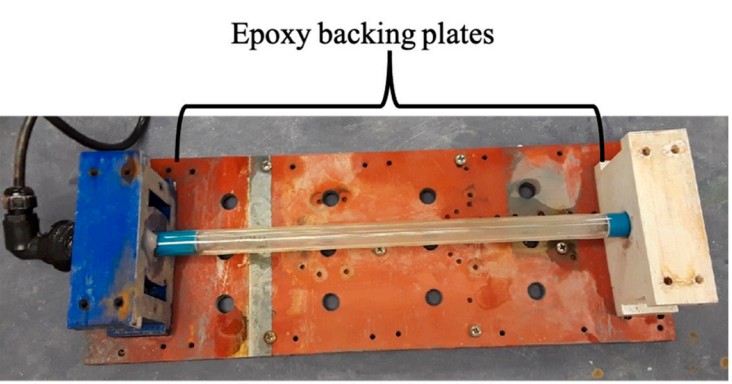

Epoxy backing plates

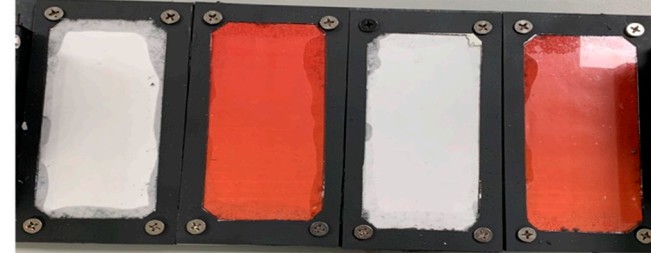

Test surfaces (10 cm X 20 cm) attached to backing plates

**Figure 1.** A visual representation of the UVC box set up. The sides of the box were composed of two epoxy backing plates (10 cm × 20 cm) with 3D printed caps that allowed for the lamp to be held in place. The photos in this figure reflect the box prior to the insertion of panels. Each box housed eight panels, four on either side, placed 25 mm from the lamp source.

**Table 1.** A summary of the experiments conducted with the number and type of substrate and the time frequency.

| Experiment | Panels | Time Frequency |
|---|---|---|
| UVC and Color | 4 Red and 4 White Polycarbonate (Box 1) | 1 min/day |
| UVC and Reflectance | 4 Red and White Polycarbonate (Box 1) 8 Stainless steel (Box 2) 4 Polycarbonate and 4 Stainless steel (Box 3) | 1 min/day |
| UVC and Exposure Intervals | 4 Polycarbonate and 4 Stainless steel | 1 min/week |
| | 4 Polycarbonate and 4 Stainless steel | 5 min/week |
| | 4 Polycarbonate and 4 Stainless steel | 10 min/week |
| | 4 Polycarbonate and 4 Stainless steel | 10 min/MF |
| | 4 Polycarbonate and 4 Stainless steel | 10 min/MWF |
| | 4 Polycarbonate and 4 Stainless steel | 10 min/day |

### 2.3. UVC and Exposure Intervals

According to the above results from Section 2.1, two separate time–frequency experiments were conducted to investigate the duration in which UVC would be the most effective on the inert surfaces. Color was not considered as a factor following the results from Section 2.1. The previous two experiments, Sections 2.1 and 2.2, tested UVC at daily intervals (Table 1), which was able to provide reduced macrofouling settlement. In order to determine if a weekly dose could also be effective, three weekly frequencies were tested for 1 month immersion (Table 1): 1 min a week, 5 min a week, and 10 min a week. Following these results, a second experiment was conducted using 10 min UVC intervals at a more frequent rate: 10 min twice a week (every Monday and Friday), 10 min three times a week (every Monday, Wednesday, and Friday), and 10 min every day.

The same box setup described above and shown in Figure 1 was used during these experiments, housing four stainless steel panels and four polycarbonate panels. Each box contained a 25 W Aqua UVC (254 nm) lamp placed in the center at a distance of 25 mm from the panels on each side of the box and connected to a PRIME-digital lighting timer to control the frequency. A set of control panels was constructed for each surface material, immersed, and not exposed to UVC.

### 2.4. Immersion

Testing was conducted at the Center for Corrosion and Biofouling Control's static immersion facility located at Port Canaveral, Florida (28°24′31.01″ N, 80°37′39.54″ W). The average salinity at this location is $34 \pm 2$ ppt, and the average water temperature is $27 \pm 2$ °C. Biofouling is high year-round, with seasonality observed with different fouling organisms. For example, in the warmer months (June, July, August), encrusting bryozoans, calcareous tubeworms, and barnacles dominate, while, in the cooler months (December, January, February), arborescent bryozoans and biofilms dominate. All panels were immersed 0.5 m below the surface water at Port Canaveral for a 1 month period [33].

Fouling coverage was visually assessed monthly. Only organisms that were directly attached to the surface were recorded [33]. For experiments that continued past 1 month of immersion, all panels were cleaned back and wiped down with vinegar, which is a mild acid that helps neutralize the surface and remove settlement cues. After cleaning, the timer was then switched to the next set of predetermined time frequencies.

### 2.5. Statistical Analysis

A permutational multivariate analysis of variance (PERMANOVA) was performed on data collected for each month, to compare the total biofouling communities based on UVC treatment. A PERMANOVA analysis was run on both UVC-exposed and nonexposed panels to compare color (red and white), surface material (polycarbonate and stainless steel), the indirect effect of UVC, and the exposure intervals. A nonmetric multidimensional scaling (MDS) plot was also conducted, when PERMANOVA results proved to be significant, to display differences in fouling communities based on surface material. A SIMPER analysis was used to indicate differences among fouling communities for the MDS plots. In addition, a multivariate analysis of variance (ANOVA) followed by a Tukey test was used to compare individual biofouling groups (i.e., barnacles, tubeworms) for different surface materials and between exposed samples and controls. All analyses were done using R statistical software (2019).

## 3. Results

### 3.1. UVC and Color

The synergistic effect of color and UVC exposure on biofouling was assessed using panels immersed in Box 1 (red and white polycarbonate) compared to control panels (no UVC). After 1 month exposure of UVC at a frequency of 1 min per day, all panels were 100% fouled. The red UVC-exposed panels were primarily fouled with biofilm and had an average of $31 \pm 21.7\%$ of macrofouling composed of tubeworms, tunicates,

encrusting bryozoan, and intermediate bryozoan (Figure 2). White UVC-exposed panels had an average of 9 ± 4.7% of macrofouling, with the rest of the community driven by biofilm. Between the red and white exposed panels, no statistical difference was found with the overall biofouling community ($p > 0.05$). However, there was a statistical difference in tubeworm settlement between the two colors ($p < 0.05$), with red having an average of 7% and white having an average of 3% tubeworm abundance. While the intermediate bryozoan visually appeared to vary between the red and white UVC-exposed panels, statistically, there was no significance ($p > 0.05$). Control panels were both 100% fouled, but red panels had 67 ± 35.5% macrofouling, while white panels had 12 ± 4.6% macrofouling (Figure 2). Community composition of red controls was primarily characterized by tubeworms with a minimal amount of tunicate settlement. Macrofouling on the white controls was mainly tubeworm settlement, with a low settlement of many other organisms such as bryozoans, barnacles, and sponge. No statistical significance was found between UVC-exposed panels and controls ($p > 0.05$). Similar to the UVC-treated panels, the controls showed no significant variance between colors for total biofouling coverage ($p > 0.05$) but did show significant difference between the red and white for tubeworm settlement with 56 ± 35.5% and 6 ± 4.6%, respectively ($p < 0.05$).

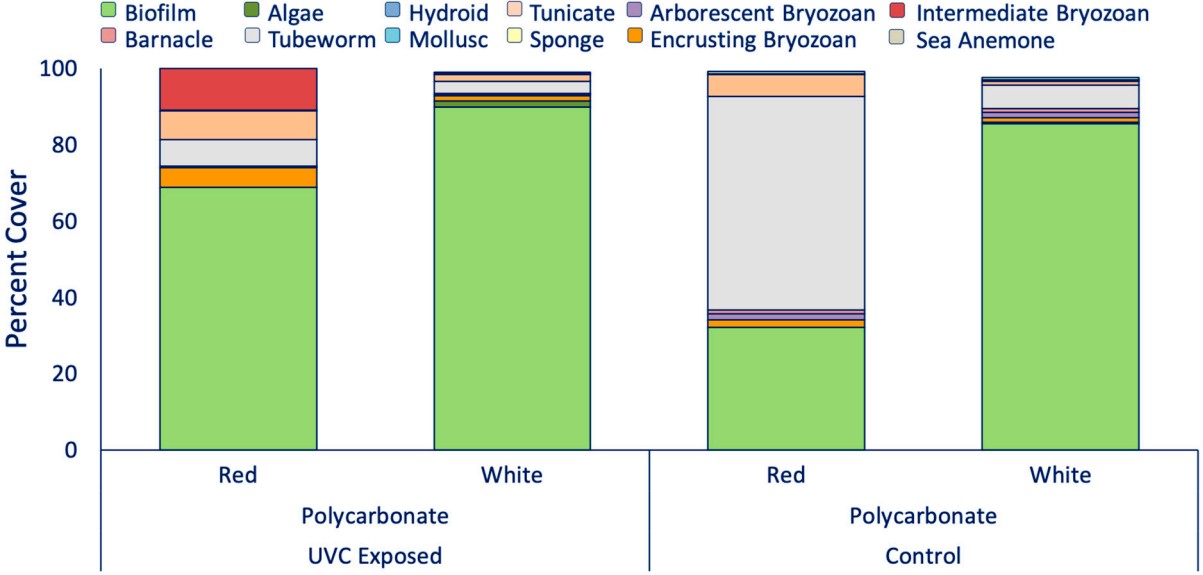

**Figure 2.** Biofouling composition on the UVC exposed red and white panels and their corresponding controls.

### 3.2. UVC and Reflectance

To address if surface material and UVC can be used in synergy to prevent biofouling, Box 1 (all polycarbonate) and Box 2 (all stainless steel) were utilized. After 1 min a day exposure, all the stainless steel and polycarbonate panels were 100% fouled (Figure 3). UVC polycarbonate panels had 20 ± 18.7% macrofouling coverage, which consisted primarily of encrusting bryozoan, tubeworm, tunicates, and intermediate bryozoan. UVC-exposed stainless steel panels had an average macrofouling coverage of 54 ± 20.7%. The community was driven by tubeworms, tunicates, and encrusting bryozoan. For both types of material, biofilm was also a major contributor to the overall fouling community (Figure 3). PERMANOVA and ANOVA results indicated a difference among stainless steel and polycarbonate biofouling communities, with significant differences for biofilm, encrusting bryozoan, and tubeworms ($p < 0.05$) (Figure 4).

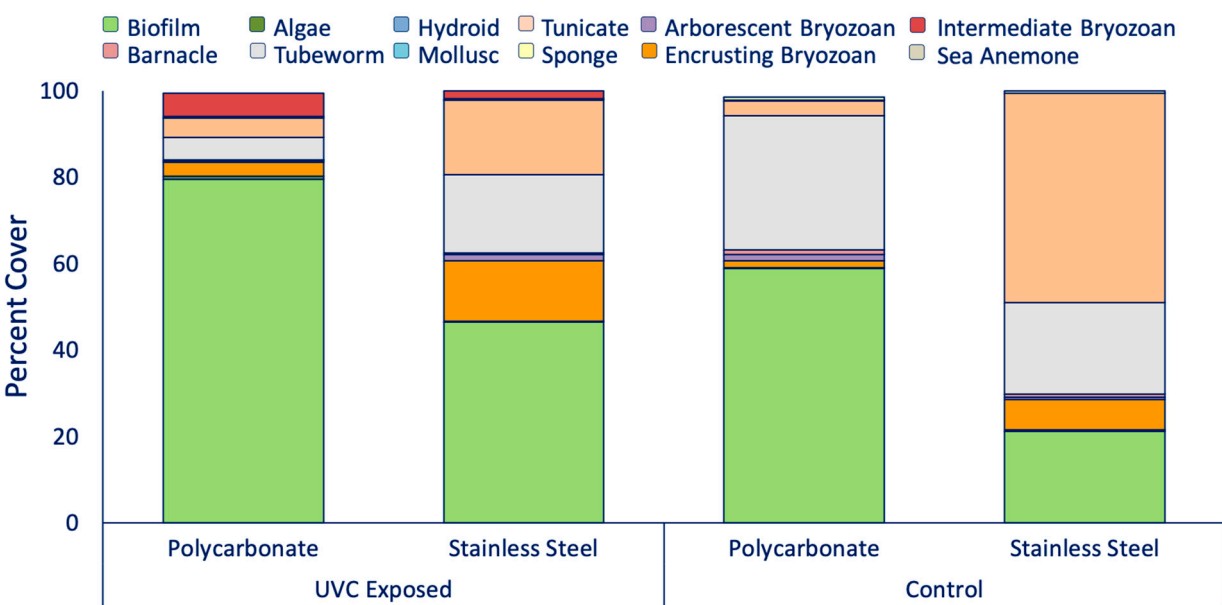

**Figure 3.** Biofouling composition on the UVC-exposed polycarbonate (average of red and white) and stainless steel panels and their corresponding controls.

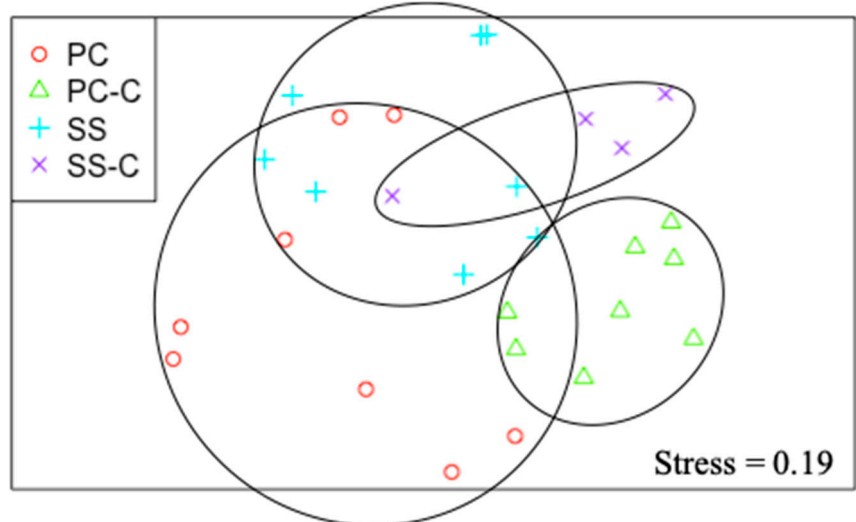

**Figure 4.** MDS of average biofouling communities on both UVC-exposed polycarbonate (PC) and stainless steel (SS) and nonexposed polycarbonate (PC-C) and stainless steel control surfaces (SS-C) following the 1 min/day exposure time.

Controls had parallel results to the UVC exposed panels, with significant biofouling community differences ($p < 0.05$) (Figure 4). Polycarbonate control panels were predominantly fouled with biofilm and $40 \pm 37.5\%$ macrofouling which consisted primarily of tubeworms. Stainless steel control panels had a greater macrofouling community presence ($79 \pm 17.9\%$) which was typified by tunicates, tubeworms, and encrusting bryozoan (Figure 3). UVC-exposed panels had significantly less biofouling abundance than untreated panels ($p < 0.001$), with greater tunicate settlement on the controls ($p < 0.001$). These trends can also be seen on the MDS plot in Figure 4.

To test if there was an indirect effect with surface material, a box (Box 3) was arranged to have both red polycarbonate and stainless steel panels placed directly across from one another. After 1 month, biofouling was similar among exposed surfaces ($p > 0.05$) (Figure 5). The macrofouling for control panels was significantly different between the

stainless steel and polycarbonate panels, with less on the polycarbonate panels ($p < 0.001$). Macrofouling on control stainless steel panels differed significantly from the UVC stainless steel panels that were exposed to indirect UVC ($p < 0.05$), and similar results were found among polycarbonate panels ($p < 0.05$). Comparisons were also made with polycarbonate and stainless steel panels from Box 1 and Box 2, respectively, where the panels were not placed directly across from one another. As described above, the total macrofouling on the directly exposed UVC polycarbonate and stainless steel panels was $20 \pm 18.7\%$ and $54 \pm 20.7\%$, respectively, comprising tubeworm, tunicates, and encrusting bryozoan. For the indirect panels, polycarbonate had $20 \pm 9.4\%$ macrofouling and stainless steel panels had $26 \pm 6.3\%$ macrofouling. ANOVA indicated that macrofouling recruitment was significantly different between those panels which received direct UVC and those which received indirect UVC, for both polycarbonate and stainless steel ($p < 0.05$). Specifically, tunicates were less abundant on the stainless steel panels in the indirect treatment, but tubeworms were greater on directly exposed stainless steel panels (Box 2).

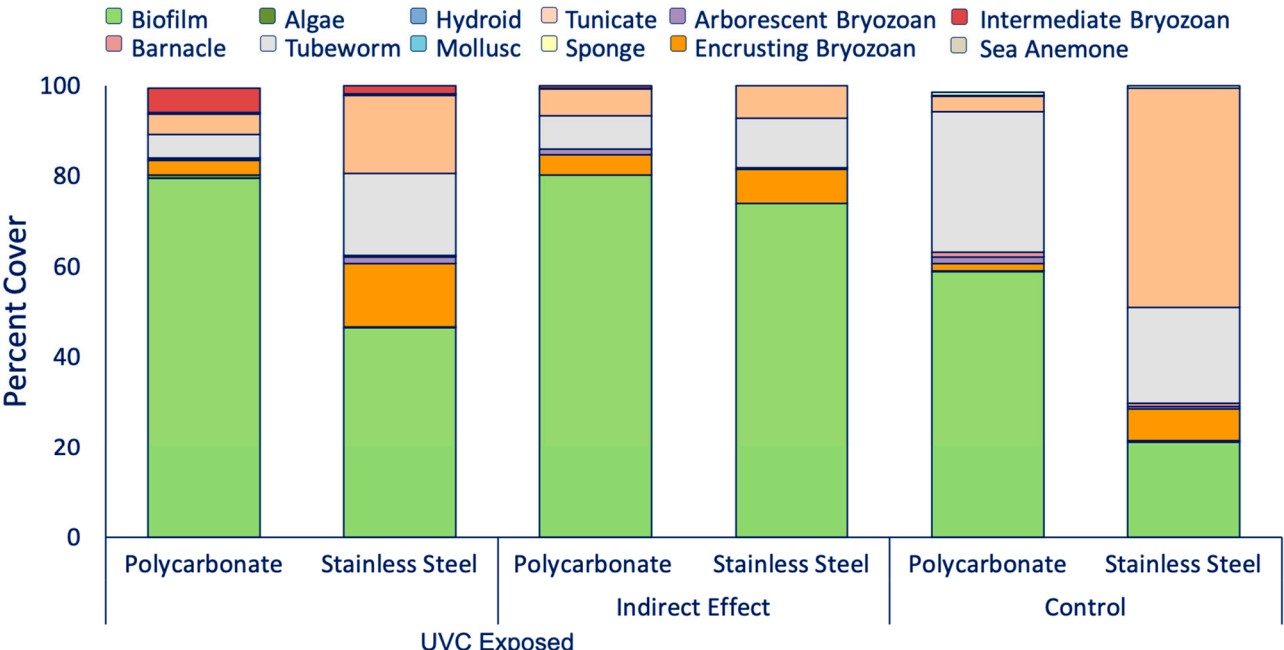

**Figure 5.** Biofouling composition on the UVC-exposed polycarbonate (red) and stainless steel panels and their corresponding controls.

### 3.3. Determination of UVC Exposure Intervals

Two exposure interval experiments were performed to evaluate the duration at which UVC may be effective in preventing biofouling, i.e., weekly and 10 min increments. During the first month, weekly time frequencies were tested (Table 1): 1 min/week, 5 min/week, and 10 min/week. During this exposure, stainless steel and polycarbonate panels were 100% fouled with biofilm and little macrofouling (Figure 6). The 1 min/week polycarbonate panels were predominantly fouled with biofilm and had $4 \pm 1.3\%$ macrofouling which was mainly tubeworms, while stainless steel panels had $23 \pm 26.1\%$ macrofouling which was primarily hydroids (~20%). When the 1 min/week panels were compared to the controls, there was a significant difference in the macrofouling community which had developed ($p < 0.05$). Both control polycarbonate and stainless steel panels had greater coverage of tubeworms (~9% and ~8%, respectively) and tunicates (~20% and 11%, respectively) than panels exposed to UVC.

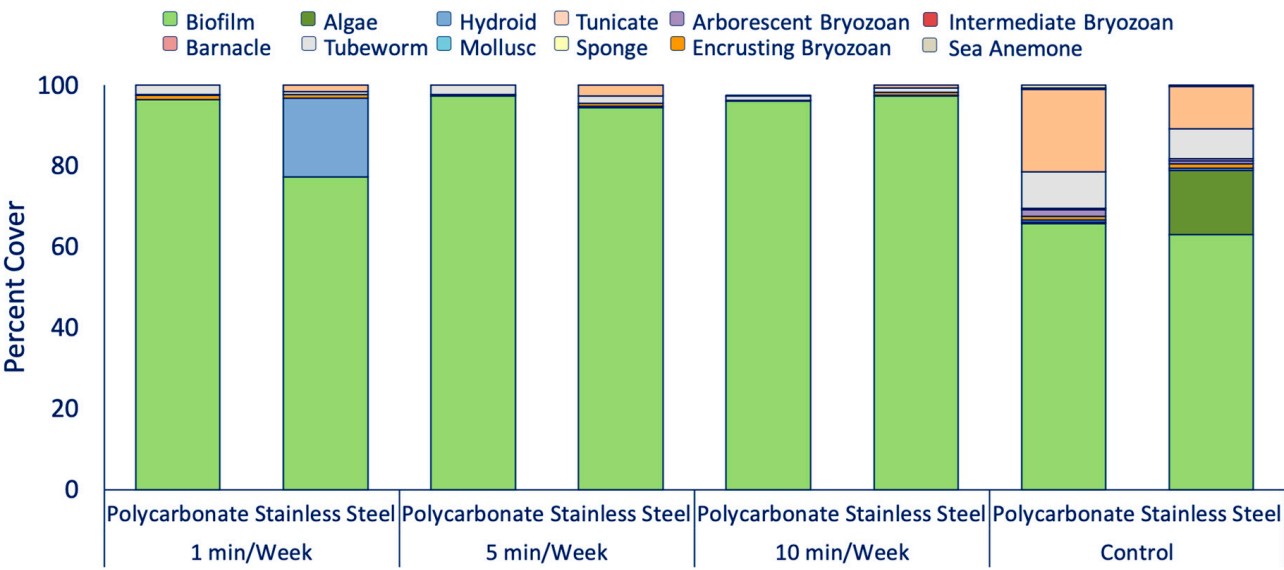

**Figure 6.** Biofouling composition on the UVC-exposed polycarbonate and stainless steel panels and their corresponding controls.

Biofouling communities did not differ between polycarbonate and stainless steel for the 5 min/week UVC-exposed panels ($p > 0.05$). This frequency had significantly less arborescent bryozoan, tubeworms, and sea anemones than the control panels ($p < 0.001$). The 10 min/week panels also had similar results to the 1 min and 5 min per week panels ($p > 0.05$). An ANOVA test was performed on all time frequencies (1 min/week, 5 min/week, and 10 min/week), but showed no statistical difference with regard to the total biofouling community ($p > 0.05$) among the three exposure treatments. UVC treatment for all frequencies reduced the macrofouling that occurred on both polycarbonate and stainless steel surfaces.

According to the first month of results, 10 min UVC exposure treatments were applied (Table 1): 10 min twice weekly (MF), 10 min three times weekly (MWF), and 10 min/day. All treatments significantly reduced the macrofouling compared to the controls, and the 10 min/day treatment prevented all macrofouling and significantly reduced biofilm (Figure 7). For the twice weekly exposed panels, polycarbonate had $45 \pm 45.4\%$ macrofouling, while the stainless steel had $74 \pm 28.6\%$ macrofouling. As for the panels exposed three times weekly, polycarbonate had $18 \pm 10.4\%$ macrofouling and stainless steel had $65 \pm 37.9\%$ macrofouling. Both the polycarbonate and the stainless steel panels exposed to UVC daily were approximately 40% covered in primarily biofilm. The stainless steel panels also had a very minimal amount macrofouling, $1 \pm 0.6\%$ (Figure 7).

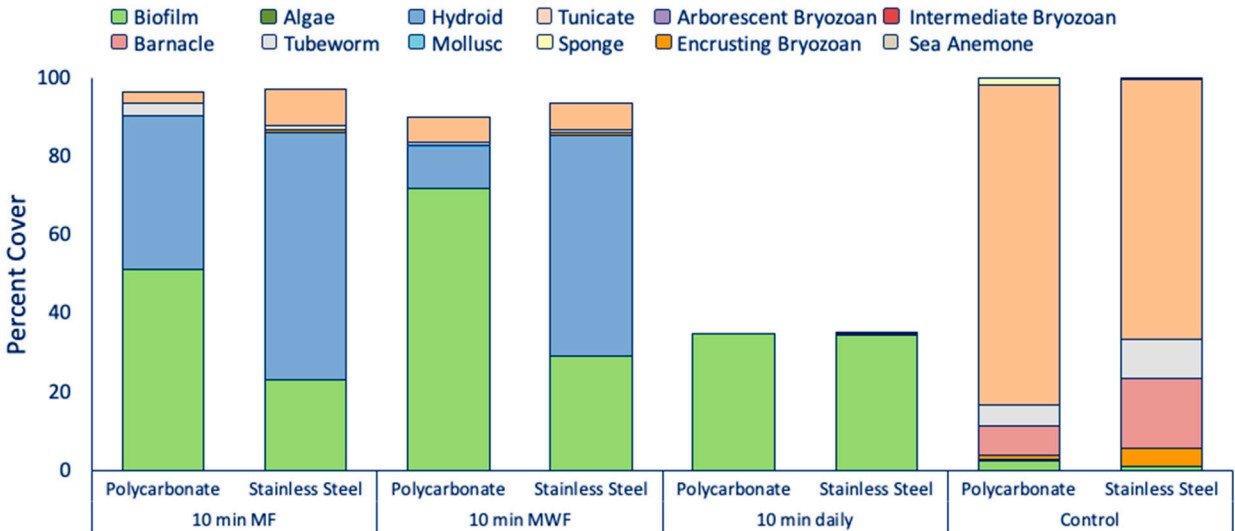

**Figure 7.** Biofouling composition on the UVC-exposed polycarbonate and stainless steel panels and their corresponding controls.

The total biofouling community and macrofouling community did not differ statistically among the polycarbonate and stainless steel panels exposed to UVC twice weekly ($p > 0.05$). However, when compared to the controls, they had significantly higher encrusting bryozoan ($p < 0.05$), barnacles ($p < 0.001$), tubeworms ($p < 0.05$), and tunicates ($p < 0.001$). The polycarbonate and stainless steel panels exposed to UVC three times a week presented with parallel results to the panels exposed twice weekly, where there was no significant difference among UVC-exposed panels ($p > 0.05$). When comparing time exposure treatments, there was a significant difference in total biofouling community and the macrofouling community with reference to surface material ($p < 0.05$) and time exposures ($p < 0.001$), which can be seen in Figure 8. SIMPER results indicated that the division between exposed surfaces and controls and among time frequencies was driven by biofilm formation.

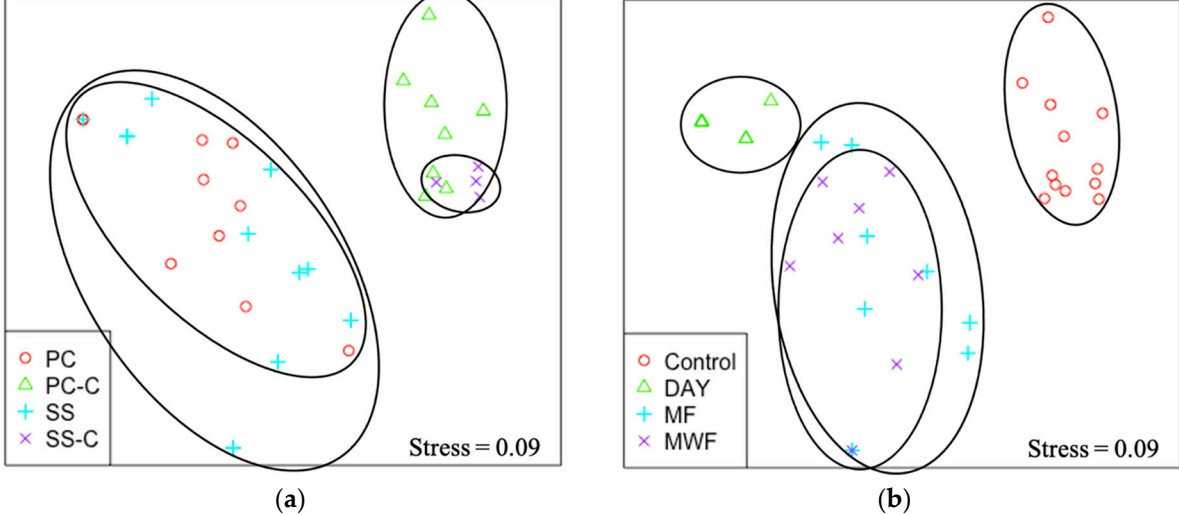

**Figure 8.** MDS of average biofouling communities among (**a**) polycarbonate (PC), stainless steel (SS), and nonexposed polycarbonate control (PC-C) and stainless steel control (SS-C), and (**b**) for the 10 min intervals: 10 min/day (Day), 10 min twice weekly (MF), 10 min three times weekly (MWF).

## 4. Discussion

UVC was applied to surfaces of different colors and two different materials, polycarbonate and stainless steel, to study the synergistic effect on biofouling prevention. This research demonstrated that UVC may be applied intermittently to various surfaces submerged in seawater as a means to control or to prevent fouling. It showed that the effectiveness is moderated by the type of surface that is being treated, and by the frequency and duration of the treatment. No significant differences in total biofouling recruitment were observed for the UVC-treated colors evaluated by these experiments. However, it was found that, in certain cases, calcareous tubeworm recruitment to the red surfaces was greater than that to the white surfaces. These results were similar to the results of Satheesh and Wesley (2010) [27], who looked at substrate color effects on the settlement of tubeworms, and Swain et al. (2006) [24], who looked at the effects of black and white surfaces on in situ fouling communities. Both studies found that fouling organisms settle on darker shades of color. Swain et al. (2006) [24] had increased settlement on black surfaces and Satheesh and Wesley (2010) [27] had greater settlement on red surfaces. Color preference is a contributing factor in biofouling recruitment and settlement for specific organisms, but it appears to be hindered with UVC exposure due to no statistical difference with respect to fouling. The UVC treatment significantly reduced the tubeworm abundance on treated panels versus the controls. As another possibility, UVC exposure to the surrounding water column could have potentially limited the settlement of tubeworms compared to the controls. However, it is unclear exactly what factor resulted in similar fouling communities.

Comparisons were made of direct UVC exposure on polycarbonate and stainless steel panels to those subjected to indirect influence. A lower abundance of biofouling was seen on both polycarbonate and stainless steel panels that were exposed indirectly. The low abundance of fouling may be due to the coupling of both surfaces reflecting and absorbing the UVC light. For example, polycarbonate reflects 0% of UVC [31], which indicates that it is absorbing the light. The absorption of light could potentially be heating the surfaces, preventing fouling from settling. The same statement could be made about stainless steel because it only reflects 50% of light [30]. The differences may also be the result of spatial variation within the test site, which could be teased out with additional replication and testing. Overall, throughout the different experiments, stainless steel had more macrofouling than the polycarbonate, which was seen specifically for tubeworms, tunicates, and encrusting bryozoan. Kim and Kang (2020), [34] saw similar results for stainless steel and polyvinyl chloride (PVC) when exposing UVC to foodborne pathogens. Bacterial colonies were less abundant on PVC (another commonly used inert surface) compared to stainless steel, indicating that reflectance may not play a role in biofouling prevention as much as the actual surface material.

While differences in color or surface material may play a small role, the major factor influencing settlement on stainless steel and polycarbonate was the duration and frequency of UVC exposure. Panels exposed for intervals of two (MF) and three times (MWF) a week resulted in biofouling communities that were similar. However, a significant decrease in fouling was seen with daily exposures of UVC. This indicated that, with a high enough dose of UVC, most if not all macrofouling can be prevented on both substrates. Hunsucker et al. (2019) [22] and Braga et al. (2020) [23] also found that longer exposure times to antifouling and fouling release coatings were effective at preventing both biofilms and macrofouling. While more work is needed to understand the impacts of UVC exposure at the different life stages of biofouling organisms, previous studies have also found veliger [35] and nematode larvae [36] to have higher rates of mortality after long exposures to UV irradiance. Both organisms were exposed to continuous UV light for longer than 24 h; however, high mortality was observed within the first 24 h. These studies correlate with what was observed when UVC was exposed to panels for 10 min a day herein. The panels exposed to UVC for 10 min a day had minimal fouling, indicating that a slight increase in time may be equivalent to continuous exposure and may display complete biofouling prevention or removal. For the current study, testing was conducted in warmer

months of Florida which are typically the more aggressive fouling season, thus requiring a longer UVC exposure. During lower fouling months or environments, a lower time and frequency of UVC exposure would be required to prevent fouling [22]. Studies comparing fouling response in different seasons will need to be further investigated since the current study was conducted in the summer months where animal larvae are the dominate fouling organisms compared to the winter months, which are more algal-dominated. Along with exposure time and intensity, distance can play a role in UVC effectiveness. Braga et al. (2020) [23] applied UVC light to pre-existing adult barnacles attached to an inert surface and saw greater mortality on panels closer to the lamp (25 mm) than farther away (275 mm).

On both polycarbonate and stainless steel, biofouling was minimal when UVC was applied for long and frequent intervals, which was also observed by Hunsucker et al. (2019) [22] and Braga et al. (2020) [23]. According to the results herein, UVC can be considered a more effective environmentally friendly use for biofouling prevention compared to fouling control coatings. Applying UVC to inert coatings or substrates could reduce the use of antifouling coatings such as copper, which can leach, and other chemicals from antifouling coatings into water bodies. UVC exposure to copper-based coatings has demonstrated that UVC may accelerate coating release [22]. Further testing is still needed to understand how long-term UVC impacts both fouling control coatings and inert surfaces. Furthermore, material selection should be considered due to the possibility of surface degradation by the UVC exposure [37].

## 5. Summary

These experiments demonstrated that low doses of UVC reduce macrofouling coverage, and its effectiveness can be improved by accurately matching it with the appropriate substrate. Both the color and the surface material experiments demonstrated that material selection influences fouling organism settlement. While inert surfaces are not typically used in the construction of naval ships, they can be used in other oceanographic fields, such as with ROVs and instrumentation. Biofouling on marine sensors is problematic because it hinders the functionality of the instrument. Being able to apply UVC to the instruments, through external exposure or embedded into the device [38,39], to prevent fouling (instead of manually removing fouling) would save time and increase longevity of the instrument. Proper UVC exposure could also prove beneficial in invasive species reduction especially as a treatment in niche areas (i.e., ballast tanks and propellers) where antifouling coatings are not typically applied [38–41]. The versatility of UVC [17,22,23,38] makes it possible to use in confined spaces such as niche areas, while it can additionally be used to treat the waterbody to eliminate larvae before they are able to reproduce.

**Author Contributions:** Conceptualization, K.N.R., K.Z.H. and G.S.; methodology, K.N.R., K.H., K.Z.H., G.S. and H.G.; formal analysis, K.N.R.; investigation, K.N.R. and K.H.; data curation, K.N.R. and K.H.; writing—original draft preparation, K.N.R. and K.Z.H.; writing—review and editing, K.H., K.Z.H., G.S., and H.G.; supervision, K.Z.H. and G.S.; project administration, K.Z.H. and G.S.; funding acquisition, K.Z.H. and G.S. All authors have read and agreed to the published version of the manuscript.

**Funding:** This research was funded by The Office of Naval Research, grant numbers N00014-16-1-3123 and N00014-20-1-2214. Support for the publication fee was provided by the OA Subvention Fund at Florida Institute of Technology.

**Institutional Review Board Statement:** Not applicable.

**Informed Consent Statement:** Not applicable.

**Data Availability Statement:** Request to corresponding author of this article.

**Acknowledgments:** The authors would like to thank the members of the Center for Corrosion and Biofouling Control for their assistance on this project, Glenn Miller for his guidance for the statistical analysis, and the OA Subvention Fund at Florida Institute of Technology for funding this publication.

**Conflicts of Interest:** The authors declare no conflict of interest.

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
