# Peer review of "The Application of UVC Used in Synergy with Surface Material to Prevent Marine Biofouling"

_jmse, doi:10.3390/jmse9060662_

Round 1
Reviewer 1 Report
The manuscript concerns the potential application of UVC to surfaces of different artificial materials, i.e. stainless steel (UV-reflective) and polycarbonate (UV-transparent) of different colours (white, i.e clear, and red, i.e. dark) to study the synergistic effect on larval recruitment and settlement with the aim to consider the eco-friendly use of UVC for biofouling prevention and also to reduce the employment of antifouling paints containing biocides, the latter representing a severe risk for the coastal environment. Both the research and the experimental plan appear to be original and interesting.
Major comments:
1) The paper considers only animal taxa since a generic term “Algae” is only reported on the legend of histograms. This is not incorrect for the biofouling impact on submerged structures but, as no comment on plants is reported throughout the text, the Authors should better develop the larval behaviour beginning from the introduction by adding, e.g., “Barnacle larvae have also been shown” (line 58), “that color would influence larval settlement” (line 74), and so. The problem of larval recruitment and settlement by different surface conditions must be better emphasized.
2) In the abstract, although the first part of results (i.e. UVC and reflectance) are well reported (“There was no variance in the biofouling communities for colored surfaces when exposed to 254 nm UVC”), the second (“Reflectance studies”) and third (“Exposure intervals”) parts are too general and vague. A better explanation is necessary for “plays a role” “with significance” and “to influence”.
3) The experimental period for all panels and treatments is not clear. Line 147: “for a two-month period”. Line 168 (and others): “one-month exposure”.
4) Figure 1. A scheme of the apparatus complete of panel arrangement inside is necessary because the photos are very difficult to interpret. The space for a draft is available over or below the horizontal photo.
5) Lines 295-296. Although the colour preference was widely demonstrated as also reported by the Authors, the results do not demonstrate the hindering of the “preference” of larval settlement after an 1 min/day exposure. This statement must be reviewed. The heating of the surfaces by UV absorption (as reported in line 303) could prevent the biofilm formation (and the consequent settlement of macrofouling species), and determine mortality of juveniles and adults already settled as reported in line 319.
6) Future applications. Note that polycarbonate exposed to UV light leads to surface degradation. This in turn affects various properties of the polymer, particularly impact strength and clarity. To reduce yellowing and loss of mechanical properties, in exterior long-term applications, protection/stabilization of polycarbonate must be considered (Tjandraatmadja GF, Burn LS, Jollands MJ, 1999. The effects of ultraviolet radiation on polycarbonate grazing. In: MA Lacasseand and DJ Vanier (eds) Durability of Building Materials and Components. National Research Council, Institute for Research in Construction, Ottawa, pp. 884-898).
Minor comments:
1) Check the numbered citations inside square brackets throughout the text (mainly in Introduction and Discussion). For example, (line 60) [17] must be added after Salters & Piola (2017), (line 63) [22] after Hunsucker et al. (2019), [23] after Braga et al. (2020), (line 71) [24] after Swain et al. (2006), and so.
2) Check in figures 2,3,5,6 “Polycarobonate” and change with “Polycarbonate”.
3) Line 228. Check “Specifcially”.
4) Check line 215, where only red polycarbonate are reported in the experiment, and the legend of Fig. 5, where both red and white panels are reported.
Reviewer 2 Report
The topic of the manuscript is interesting and relevant to aims and scopes of respectable Journal of Marine Science and Engineering. The paper presents the possibility for the application of UVC along with surface material to prevent marine biofouling. The manuscript is well written. However, conclusion section must be provided within the manuscript, which would summarize the main finding of this study. Additionally, I have some comments which can improve the quality of the paper:
1) Introduction
The novelty of the work must be clearly addressed and discussed, compare your research with existing research findings and highlight novelty.
2) The last paragraph in the introduction should be rewritten. This paragraph can be moved into materials and methods section. The main objective of the work must be written on the clearer way at the end of introduction section.
3) Materials and methods
How many panels were tested in total and per case study, i.e. within UVC and Color, UVC and Reflectance and UVC and Exposure Intervals?
4) Materials and methods
One table or figure which summarises the performed experiments should be given within this section.
5) The authors should comment about applying low doses of UVC in Part 1 and Part 2 of their investigation. How long and at which frequency is UVC applied within these experiments? Authors should comment whether different results may be expected if the UVC light was applied more frequent or longer.
6) The manuscript is missing conclusion section, which should summarise the main findings of this study.
7) Within conclusion section, the authors should provide some perspective related to the future research work.
